# A Cooperative Multi-Agent Framework for Zero-Shot Named Entity Recognition

Submission Id: 812

## Abstract

Zero-shot named entity recognition (NER) aims to develop entity recognition systems from unannotated text corpora. This task presents substantial challenges due to minimal human intervention. Recent work has adapted large language models (LLMs) for zero-shot NER by crafting specialized prompt templates. And it advances the models' self-learning ability by incorporating self-annotated demonstrations. Two important challenges persist: (i) Correlations between contexts surrounding entities are overlooked, leading to wrong type predictions or entity omissions. (ii) The indiscriminate use of task demonstrations, retrieved through shallow similarity-based strategies, severely misleads the inferences made by LLMs.

In this paper, we introduce CMAS, viz., *cooperative multi-agent system*, a framework for zero-shot NER that uses the collective intelligence and tailored abilities of multiple agents to address the challenges outlined above. Cooperative multi-agent system (CMAS) has four main agents: (i) a self-annotator, (ii) a type-related feature (TRF) extractor, (iii) a demonstration discriminator, and (iv) an overall predictor. To explicitly capture correlations between contexts surrounding entities, CMAS reformulates NER into two subtasks: recognizing named entities and identifying entity type-related features within the target sentence. Moreover, pseudo-labels for TRFs are generated using mutual-information criteria without requiring human effort, facilitating the prediction of the TRF extractor. To assess the quality of demonstrations, a demonstration discriminator is established to incorporate the self-reflection mechanism, automatically evaluating helpfulness scores for the target sentence and enabling controllable utilization of demonstrations.

Experimental results show that CMAS significantly improves zero-shot NER performance across six benchmarks, including both domain-specific and general-domain scenarios. Furthermore, CMAS demonstrates its effectiveness in few-shot settings and with various LLM backbones.[1]

## Keywords

Information extraction, Named entity recognition, Zero-shot learning, Large language models, Multi-agent system

---

[1]Our code is available at https://anonymous.4open.science/r/WWW25-CMAS-8F0F.

**ACM Reference Format:**
Anonymous Author(s). 2025. A Cooperative Multi-Agent Framework for Zero-Shot Named Entity Recognition. In *Proceedings of the ACM Web Conference 2025 (WWW '25), 28 April – 2 May, 2025, Sydney, Australia.* ACM, New York, NY, USA, 13 pages. https://doi.org/XXXXXXX.XXXXXXX

## 1 Introduction

In named entity recognition (NER), the task is to identify predefined named entities, such as persons, locations, and organizations, based on their contextual semantics within input texts. NER serves as a fundamental task in information extraction and is integral to various downstream natural language processing (NLP) applications, including question answering [17, 37], document understanding [12], and information retrieval [5, 6]. Current NER methods primarily use fully supervised learning paradigms and show impressive performance across various benchmarks [30, 39, 51]. However, these fully-supervised paradigms rely heavily on large-scale, human-annotated data. In real-world scenarios, the availability of annotated data may be restricted to specific domains, severely hindering the generalizability and adaptability of NER models [47, 60].

Recently, large language models (LLMs) have transformed natural language processing with their zero-shot or few-shot generalization abilities [3, 24, 26, 27]. With their extensive search spaces and large-scale pre-training data, LLMs have the potential to overcome the challenges of data sparsity and generalizability faced by NER models. Motivated by these developments, one line of prior work explores prompting techniques to enhance few-shot in-context learning (ICL) for NER [45]. Other efforts use specialized knowledge to develop task-specific LLMs for NER [35, 60, 62] or employ LLMs as data annotators or generators to augment smaller language models [14, 22]. However, these approaches still require deliberately selected annotated task examples or external knowledge. The reasoning abilities of LLMs for zero-shot NER remain underexplored.

To address zero-shot NER, Wei et al. [49] transform this task, where no labeled data is available, into a two-stage question-answering process by chatting with LLMs. Xie et al. [54] conduct a systematic empirical study on zero-shot NER using LLMs and tailor prevalent reasoning methods, such as tool augmentation and majority voting, to adapt ChatGPT for zero-shot NER. Furthermore, to reduce reliance on external tools, Xie et al. [55] enhance the self-learning capabilities of LLMs through a self-improvement mechanism. Specifically, LLMs initially annotate unlabeled corpora with self-consistency scores. Subsequently, for each test input, inference is conducted using in-context learning (ICL) with demonstrations retrieved from the self-annotated dataset.

Despite these advances, current zero-shot NER methods still encounter two challenging problems:

**Challenge 1: Overlooking correlations between contexts surrounding entities.** Prior work [49, 54, 55] focuses exclusively on

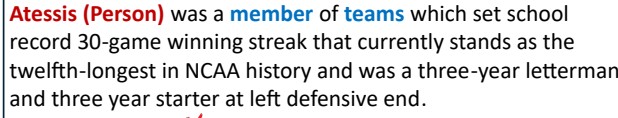

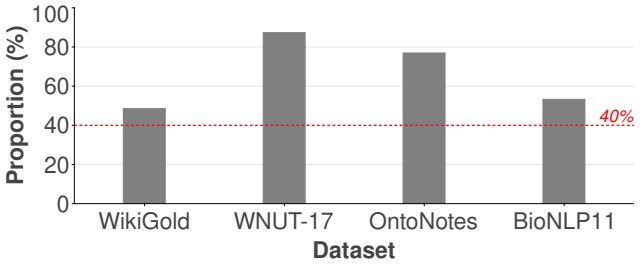

(a) Overlooking correlated contexts surrounding entities.

(b) Proportions of demonstrations without target entity types.

**Figure 1: (a) Examples of incorrect type prediction and entity omission from an existing method [54] due to overlooking correlated contexts surrounding entities. Red texts represent wrongly recognized entities; golden labels are included in the brackets. Blue texts highlight entity type-related features (TRFs), i.e., contexts strongly associated with the entity types. (b) Proportions of selected demonstrations lacking target entity types in the WikiGold [2], WNUT-17 [4], OntoNotes,[2] and BioNLP11 [31] datasets. More than 40% of demonstrations do not contain any entity types within the target sentence.**

recognizing entities within the target sentence. However, the contexts surrounding entities of the same type are correlated, and identifying contexts that are strongly associated with entity types plays a crucial role in facilitating the generalization of pretrained language models for the NER task [47]. Neglecting these contextual correlations can lead to incorrect type predictions or entity omissions, severely impeding the adaptation of LLMs to zero-shot scenarios. For instance, as shown in Figure 1(a), the existing method [54] fails to recognize "member" and "teams," which are closely related to Person entities, in the target sentence, resulting in the omission of the entity "Atessis." To tackle this issue, we propose redefining the traditional NER task into two subtasks: recognizing named entities and identifying entity *type-related features* (TRFs [47], i.e., tokens strongly associated with entity types).

**Challenge 2: Indiscriminate use of task demonstrations.** To enhance task understanding and guide inference, recent zero-shot NER methods [54, 55] use both task instructions and task-specific demonstrations as input prompts for LLMs. However, these methods employ shallow strategies for demonstration retrieval, such as random sampling and $k$-nearest neighbors, resulting in the frequent emergence of low-quality demonstrations. For instance, as illustrated in Figure 1(b), approximately 87.33% and 76.94% of selected demonstrations do not contain any target entity types in the test sentences from the WNUT-17 and OntoNotes datasets, respectively. The indiscriminate use of unhelpful or irrelevant demonstrations substantially misleads the inference process of LLMs and degrades models' prediction abilities. To address this problem, we incorporate a self-reflection mechanism [1, 38], enabling LLMs to reflect on the helpfulness of retrieved demonstrations and selectively learn from them.

**Contributions.** Note that existing LLMs suffer from severe performance degradation with long input contexts [13, 21] and complex instruction following [33, 34]. Thus, it is challenging to effectively capture contextual correlations and discriminative use retrieved demonstrations through single-turn or multi-turn dialogues with LLMs. Inspired by the demonstrated complex problem-solving capabilities of multi-agent approaches [7, 52], in this paper, we present a framework, named the *cooperative multi-agent system* (CMAS)

for zero-shot NER, harnessing the collective intelligence of LLM-based agents to address the challenges listed. As Figure 2 illustrates, CMAS consists of four main agents: (i) a self-annotator, (ii) a type-related feature extractor, (iii) a demonstration discriminator, and (iv) an overall predictor. First, adopting the self-improving strategy outlined in [55], CMAS employs an LLM as the self-annotator to create self-annotated data through predictions on unlabeled corpora. Then, to empower the simultaneous extraction of entities and contextual correlations, CMAS redefines the NER task into two subtasks: recognizing named entities and identifying entity type-related features within the target sentence. To achieve this, an LLM-based type-related feature extractor is developed to pinpoint words or phrases closely related to different entity types from the surrounding contexts using specialized in-context learning (ICL). Additionally, pseudo-labels for TRFs are generated using mutual information criteria, streamlining the inference process of the TRF extractor without requiring human interventions. Given the entity type features relevant to the target sentence, the demonstration discriminator integrates a self-reflection mechanism [1] to automatically assess the helpfulness of each selected demonstration in making predictions on the target test sentences. Finally, with the extracted TRFs and predicted helpfulness scores of demonstrations, the overall predictor performs inference on each incoming test sentence with a two-stage self-consistency strategy [46, 54], selectively learning from retrieved demonstrations while considering contextual correlations. Additionally, external tools, such as a syntactic structure analyzer [10], can be used to further enhance CMAS (see Section 7.1).

Our contributions are summarized as follows: (i) To the best of our knowledge, ours is the first study to design a cooperative multi-agent system for zero-shot NER that harnesses the collaborations and unique roles of multiple agents to integrate contextual correlations and the self-reflection mechanism. (ii) We redefine NER into two subtasks: recognizing named entities and identifying entity type-related features. In this way, CMAS effectively captures contextual correlations between the contexts during entity recognition, thereby reducing incorrect type predictions and entity omissions. (iii) We incorporate a self-reflection mechanism into the demonstration discriminator. By evaluating the helpfulness scores for entity

---

[2]https://catalog.ldc.upenn.edu/LDC2013T19

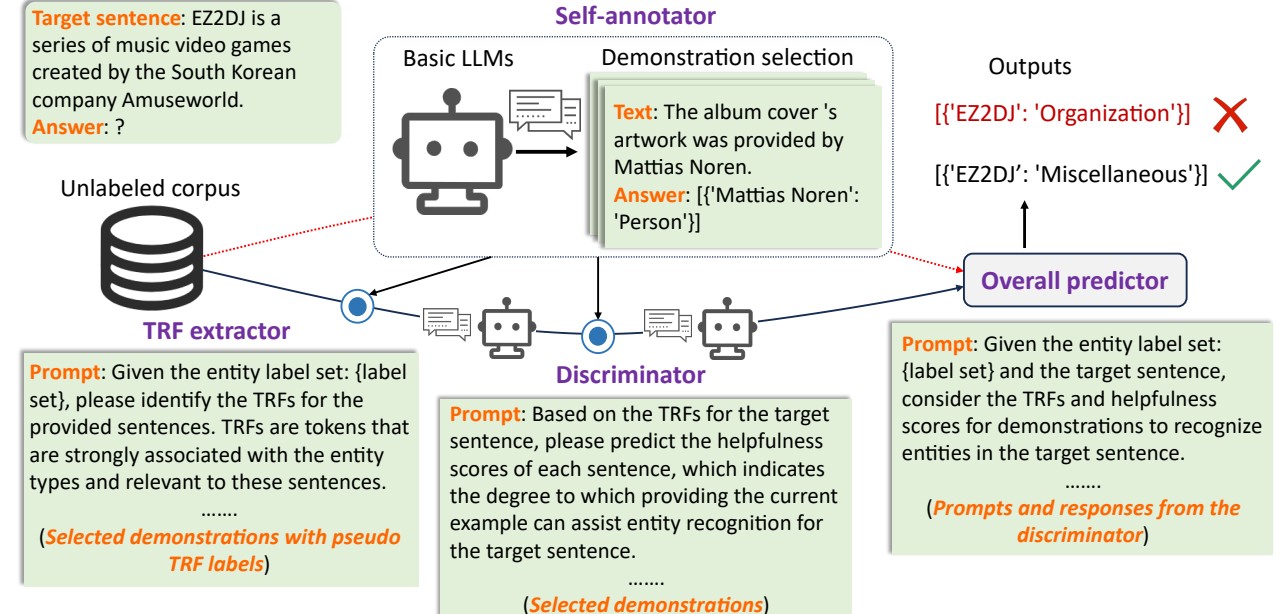

**Figure 2: An overview of CMAS. The dotted red lines indicate the workflow of an existing method [55], which leads to incorrect predictions. In contrast, the solid black lines illustrate the workflow of the proposed CMAS, which consists of four key agents: (i) a self-annotator, (ii) a type-related feature extractor, (iii) a demonstration discriminator, and (iv) an overall predictor.**

extractions in target sentences, CMAS is capable of discriminately using and learning from selected demonstrations. (iv) Experimental results across six benchmarks demonstrate that our proposed CMAS achieves state-of-the-art performance on zero-shot NER and exhibits strong robustness across varying numbers of task demonstrations.

## 2 Related Work

We investigate related work in three areas: (i) reasoning with LLMs, (ii) LLMs for IE, and (iii) LLM-based multi-agent systems.

### 2.1 Reasoning with LLMs

LLMs demonstrate strong zero-shot and few-shot reasoning capabilities, particularly when prompted to provide intermediate rationales for solving problems. Recent studies in both few-shot and zero-shot frameworks explore eliciting a chain-of-thought (CoT) process from LLMs. These studies encourage LLMs to refine their responses incrementally, enhancing the reasoning process step-by-step [15, 41, 48, 59]. Additionally, strategies like problem decomposition such as least-to-most prompting [61], break down complex problems into manageable sub-problems, addressing them sequentially. The self-consistency approach [46] involves generating a diverse array of responses from an LLM, subsequently selecting the optimal answer by averaging over these possibilities. In this paper, we focus on investigating the zero-shot reasoning ability of LLM on the NER task.

### 2.2 LLMs for IE

Evaluating the performance of LLMs on IE tasks is garnering significant attention [16, 18, 22, 45, 51, 60]. Wei et al. [49] propose a two-stage chatting paradigm for IE. In the first stage, ChatGPT is tasked with recognizing types of elements. In the second stage, it extracts mentions corresponding to each identified type. Han et al. [9] present a comprehensive analysis of LLMs' capabilities in IE tasks, examining aspects such as performance, evaluation criteria, robustness, and prevalent errors.

Xie et al. [54] conduct a systematic empirical study on the reasoning abilities of LLMs in IE, particularly examining performance in zero-shot NER tasks. Xie et al. [55] focus on enhancing the performance of zero-shot NER using LLMs by introducing a training-free self-improving framework that uses an unlabeled corpus to stimulate the self-learning capabilities of LLMs. Wan et al. [40] employ the chain-of-thought (CoT) approach for relation extraction (RE), using LLMs to generate intermediate rationales based on demonstrations from the training set.

### 2.3 LLM-based multi-agent systems

LLMs exhibit useful capabilities in reasoning and planning, aligning with human expectations for autonomous agents capable of perceiving their environments, making decisions, and taking responsive actions [8, 20, 50, 52]. Consequently, LLM-based agents are increasingly designed to understand and generate human-like instructions, enhancing complex interactions and decision-making across various contexts [19, 38, 57]. Building on the abilities of individual LLM-based agents, the concept of LLM-based multi-agent systems has been introduced. Such systems use the collective intelligence and specialized skills of multiple agents, enabling collaborative engagement in planning, decision-making, and discussions, mirroring the cooperative nature of human teamwork in problem-solving.

Recent research demonstrates the efficacy of LLM-based agents in diverse applications, including game simulation [44, 56], software development [11, 32], society simulation [28, 29], multi-robot systems [23, 58], and policy simulation [53]. Comprehensive updates on advances in LLM-based agents are detailed in recent surveys [7, 43, 52]. To the best of our knowledge, ours is the first study to develop an LLM-based cooperative multi-agent system tailored for zero-shot NER tasks.

In this paper, we focus on the zero-shot NER task. The work most closely related to ours is [49, 54, 55]. However, existing methods still face two challenging problems: (i) they overlook correlations between contexts surrounding entities, and (ii) they make indiscriminate use of task demonstrations. In our proposed CMAS, to explicitly model contextual correlations within target sentences, both named entities and TRFs are simultaneously extracted using specialized ICL. To enable controllable usage of demonstrations, a self-reflection mechanism is incorporated to automatically predict the helpfulness score of each selected demonstration for inference on the target sentences.

## 3 Task Formulation

To explicitly capture contextual correlations during the entity extraction process, we reinterpret the original NER task as two subtasks: recognizing named entities and identifying entity type-related features within the target sentence.

**Zero-Shot NER.** In this paper, we focus on the NER task in the strict zero-shot setting [55]. In this setting, no annotated data is available; instead, we only have access to an unlabeled corpus $\mathcal{D}_u$. Specifically, given an input sentence $\mathbf{x} = x_1, \ldots, x_n$ with $n$ words from the test set $\mathcal{D}_t$, our aim is to recognize structured outputs $\mathbf{y}$ from $\mathbf{x}$, consisting of a set of $(e, t)$ pairs. Here, $e$ is an entity span, a sequence of tokens from $\mathbf{x}$, and $t$ is the corresponding entity type from a predefined set $\mathcal{T}$, such as persons, location, or miscellaneous.

**TRF extraction.** Type-related features (TRFs), which are tokens strongly associated with entity types, are critical for the generalization of NER models [47]. Given an input sentence $\mathbf{x} \in \mathcal{D}_t$ including entity types $\mathcal{T}_x$, the goal of TRF extraction is to identify all TRFs $\mathcal{R}$ that are related to the input sentence $\mathbf{x}$ for all entity types in $\mathcal{T}_x$. Each TRF $\mathbf{w} \in \mathcal{R}$ is an $m$-gram span. For instance, as illustrated in Figure 1(a), "member" and "teams" are TRFs associated with the Person entity type, while "music video games" is recognized as a TRF for the Miscellaneous type.

## 4 CMAS: A Cooperative Multi-Agent System

In this section, we present the four main agents of the proposed CMAS as described in Figure 2: (i) a self-annotator (see Section 4.1), (ii) a type-related feature extractor (see Section 4.2), (iii) a demonstration discriminator (see Section 4.3), and (iv) an overall predictor (see Section 4.4).

First, the self-annotator uses an LLM to produce self-annotated data by making predictions on the unlabeled corpus and preliminarily retrieves demonstrations using a similarity-based strategy. Next, the type-related feature extractor automatically acquires pseudo-labels for TRFs using mutual information criteria and identifies words or phrases strongly associated with different entity types using specialized ICL. Subsequently, the demonstration discriminator

incorporates a self-reflection mechanism to evaluate the helpfulness scores of each retrieved demonstration for predictions on the target input sentence. Finally, given the extracted TRFs and predicted helpfulness scores, the overall predictor performs inference on the target sentences by employing question-answering prompts and a two stage self-consistency strategy.

### 4.1 Self-annotator for unlabelled data

As mentioned in Section 1 and 3, we only have access to an unlabeled corpus $\mathcal{D}_u$ in zero-shot NER. Inspired by the self-improvement strategy [55], we specify an LLM-based self-annotator to guide the inference of LLMs, which first annotates the unlabeled corpus with zero-shot prompting and then preliminarily selects demonstrations from the self-annotated data for each target sentence.

**Self-annotation.** For each unlabeled sample $x_i \in \mathcal{D}_u$, we generate predictions using LLMs with zero-shot prompting. This process is formulated as follows:

$$\mathbf{y}_i = \arg\max_{\mathbf{y}} P_s(\mathbf{y}|\mathbf{T}_s, \mathbf{x}_i), \tag{1}$$

where $\mathbf{T}_s$ is the prompt template used for self-annotation. Prompt 1 (in the Appendix) illustrates an instance of $\mathbf{T}_s$. The predictions $y_i = \{(e_i^j, t_i^j)\}_{j=1}^l$ consist of pairs of entity mentions and types, where $l$ is the number of the predicted entity mentions. $P_s$ is the output probability of the self-annotator. To enhance the reliability of the annotations, we use self-consistency [46] and adopt a *two-stage majority voting* strategy [54]. We sample multiple responses from the model. In the first stage, we consider a candidate mention as an entity if it is present in more than half of all responses; otherwise, we discard it. In the second stage, for each mention retained from the first stage, we determine the entity type label based on the majority agreement among the responses and assign this as the final predicted label.

**Preliminary demonstration selection.** When a target sentence $\mathbf{x}^q$ arrives, our goal is to retrieve $k$ relevant demonstrations $\mathbf{S} = \{\mathbf{x}_i, \mathbf{y}_i\}_{i=1}^k$ from the reliable self-annotated dataset. To achieve a better trade-off between similarity, diversity, and reliability, we employ a *diverse nearest neighbors with self-consistency ranking* strategy [55], which first retrieves $K$ nearest neighbors based on cosine similarities between sentence representations and then selects samples with the top-$k$ self-consistency scores.

### 4.2 Type-related feature extractor

To capture correlations between contexts surrounding entities, we design an LLM-based type-related feature (TRF) extractor using specialized in-context learning (ICL). We use mutual information criteria [47] to generate pseudo TRF labels $\{\mathcal{R}_i\}_{i=1}^k$ for self-annotated demonstrations $\mathcal{S} = \{\mathbf{x}_i, \mathbf{y}_i\}_{i=1}^k$, which are selected for the test input $\mathbf{x}^q$. Building on this, we apply the specialized ICL prompts to identify relevant TRFs $\mathcal{R}^q$ for $\mathbf{x}^q$.

**Pseudo-label generation.** To facilitate TRF extraction for the target sentence $\mathbf{x}^q$, we generate pseudo TRF labels for its self-annotated demonstrations $\mathcal{S} = \{\mathbf{x}_i, \mathbf{y}_i\}_{i=1}^k$ using a mutual information-based method [47]. We define $\mathcal{D}_t$ as the set containing all sentences from the unlabeled corpus $\mathcal{D}_u$ where entities of the $t$-th type appear. The complementary set, $\mathcal{D}_u \backslash \mathcal{D}_t$, includes sentences

that do not contain entities of the $t$-th type. To identify TRFs $\mathcal{R}^t$ associated with the $t$-th entity type, we apply the following filtering condition:

$$\frac{C_{\mathcal{D}_u \backslash \mathcal{D}_t}(\mathbf{w})}{C_{\mathcal{D}_t}(\mathbf{w})} \leq \rho, \quad C_{\mathcal{D}_t}(\mathbf{w}) > 0, \tag{2}$$

where $C_{\mathcal{D}_t}(\mathbf{w})$ represents the count of the m-gram $\mathbf{w}$ in $\mathcal{D}_t$, and $C_{\mathcal{D}_u \backslash \mathcal{D}_t}(\mathbf{w})$ represents its count in the rest of $\mathcal{D}_u$ excluding $\mathcal{D}_t$. The parameter $\rho$ is an m-gram frequency ratio hyperparameter. By applying this criterion, we ensure that $\mathbf{w}$ is considered a part of the TRF set of $\mathcal{D}_t$ only if its frequency in $\mathcal{D}_t$ is significantly higher than its frequency in other sets ($\mathcal{D}_u \backslash \mathcal{D}_t$). Given the smaller size of $\mathcal{D}_t$ relative to $\mathcal{D}_u \backslash \mathcal{D}_t$, we select $\rho \geq 1$ but avoid excessively high values to include features associated with $\mathcal{D}_t$ and potentially relevant to other entity types within the TRF set of $\mathcal{D}_t$. Based on this, for every self-annotated demonstration $\mathbf{x}_i \in \mathcal{S}$, we compute the Euclidean distance between BERT-based embeddings of each extracted TRF $\mathbf{w}$ and each token in $\mathbf{x}_i$, selecting the top-5 features as pseudo TRF labels $\mathcal{R}_i$ of $\mathbf{x}_i$.

**TRF Extraction.** Given the target sentence $\mathbf{x}^q$ and its corresponding demonstrations $\mathcal{S}_d = \{\mathbf{x}_i, \mathbf{y}_i, \mathcal{R}_i\}_{i=1}^k$ equipped with pseudo-labels, we construct specialized ICL prompts to facilitate the identification of relevant TRFs for $\mathbf{x}^q$. The inference process is formulated as:

$$\mathcal{R}^q = \arg\max_{\mathcal{R}} P_e(\mathcal{R}|\mathbf{T}_e, \mathcal{S}_d, \mathbf{x}^q), \tag{3}$$

where $\mathbf{T}_e$ represents the specialized ICL prompt template. Prompt 2 (in the Appendix) provides a detailed instance of $\mathbf{T}_e$. $P_e(\cdot)$ represents the output probability of the TRF extractor.

### 4.3 Demonstration discriminator

As mentioned in Section 1, demonstrations retrieved using shallow similarity-based strategies can be highly irrelevant to target sentences, severely misleading the predictions of LLMs. To address this issue, we employ an LLM-based demonstration discriminator with a self-reflection mechanism [1, 38] to automatically evaluate the helpfulness of each initially selected demonstration for making predictions on the target test sentences. To achieve this, we consider the TRFs of both the demonstrations and the target sentence, extracted by the TRF extractor (see Section 4.2), as well as the self-labeled entity labels from the self-annotator (see Section 4.1). The prompts used for helpfulness score prediction are shown in detail in Prompt 3 (in the Appendix). Formally, given demonstrations $\mathcal{S}_d = \{\mathbf{x}_i, \mathbf{y}_i, \mathcal{R}_i\}_{i=1}^k$ selected for the target sentence $\mathbf{x}^q$ with extracted TRFs $\mathcal{R}^q$, the corresponding helpfulness scores $\{h_i\}_{i=1}^k$ are predicted by

$$h_i = \arg\max_h P_d(h|\mathbf{T}_d, \mathcal{S}_d, \mathcal{R}^q, \mathbf{x}_i, \mathbf{x}^q), \tag{4}$$

where $\mathbf{T}_d$ denotes the prompt template used in the demonstration discriminator. $P_d(\cdot)$ denotes the output probability of the demonstration discriminator.

### 4.4 Overall predictor

Finally, given the extracted TRFs and predicted helpfulness scores, we establish an LLM-based overall predictor to conduct inference on the target sentences. Let $\mathcal{S}_o = \{\mathbf{x}_i, \mathbf{y}_i, h_i, \mathcal{R}_i\}_{i=1}^k$ represent the self-annotated demonstrations selected for the test input $\mathbf{x}^q$, along with

the corresponding helpfulness scores $\{h_i\}_{i=1}^k$ and TRFs $\{\mathcal{R}_i\}_{i=1}^k$. As Prompt 4 (in the Appendix) shows, CMAS conducts overall predictions on $\mathbf{x}^q$ by integrating the dialogue in the demonstration discriminator and constructing a question-answering prompt template $\mathbf{T}_o$. The overall prediction is obtained by

$$\mathbf{y}^q = \arg\max_{\mathbf{y}} P_o(\mathbf{y}|\mathbf{T}_o, \mathcal{S}_o, \mathbf{x}^q), \tag{5}$$

where $P_o(\cdot)$ denotes the output probability of the overall predictor. Similar to the self-annotation process (see Section 4.1), to improve the reliability and consistency of the final results, we sample multiple responses from LLMs and adopt a two-stage self-consistency strategy.

Now that we have described the four specialized agents that make up the core of CMAS, we recall the coordinated workflow of CMAS, as illustrated in Figure 2. To start, the self-annotator incorporates a self-improvement strategy and employs an LLM to generate self-annotated data from an unlabeled corpus $\mathcal{D}_u$, and preliminarily retrieves reliable demonstrations $\mathbf{S} = \{\mathbf{x}_i, \mathbf{y}_i\}_{i=1}^k$ for each test input $\mathbf{x}^q$. Next, the type-related feature extractor uses mutual information criteria to derive pseudo TRF labels $\{\mathcal{R}_i\}_{i=1}^k$ for the self-annotated demonstrations and facilitates identifying relevant contextual correlations $\mathcal{R}^q$ for $\mathbf{x}^q$ using specialized ICL. Furthermore, the demonstration discriminator, considering the TRFs of both the target sentence and its self-annotated demonstrations, applies a self-reflection mechanism to automatically assess the helpfulness scores $\{h_i\}_{i=1}^k$ of selected demonstrations in making predictions for each target sentence $\mathbf{x}^q$. Finally, the overall predictor constructs a question-answering prompt template $\mathbf{T}_o$, synthesizing TRFs and helpfulness scores, to obtain the final prediction $\mathbf{y}^q$ for each target sentence $\mathbf{x}^q$. Through their specialized abilities and communications, the designed agents work collaboratively to enhance both the effectiveness and generalizability of our proposed CMAS for zero-shot and few-shot NER tasks (see Section 6 and 7).

## 5 Experiments

**Research questions.** We aim to answer the following research questions: (RQ1) Does CMAS outperform state-of-the-art methods on the zero-shot NER task? (See Section 6.1) (RQ2) Can CMAS be generalized to the few-shot setting? (See Section 6.2)

**Datasets.** In our experiments, we evaluate CMAS on both general-domain and domain-specific datasets. Detailed statistics of the datasets used are shown in Prompt 5 (in the Appendix). For our general-domain experiments, we consider four commonly used benchmarks, namely the CoNLL03 [36], WikiGold [2], WNUT-17 [4], and OntoNotes[3] datasets. For our domain-specific experiments, we evaluate CMAS on the GENIA [25] and BioNLP11 [31] datasets in the biomedical domain.

For zero-shot NER, to keep API usage costs under control, we randomly sample 300 examples from the test set three times and calculate the average performance. An exception is made for WikiGold, which has a test set of only 247 examples. Additionally, we randomly sample 500 examples from the training set as the unlabeled corpus. In the few-shot settings, we examine configurations including 0-shot, 3-shot, 5-shot, and 10-shot, where "shot" refers to the

---

[3]https://catalog.ldc.upenn.edu/LDC2013T19

number of demonstrations with gold labels provided to LLMs (see Section 6.2). For each setting, we randomly sample three sets of demonstrations and compute the average results.

**Baselines** For the zero-shot and few-shot NER tasks, we compare CMAS with the following baselines: (i) **Vanilla** [54, 55] employs a straightforward and common prompting strategy that directly asks LLMs to extract entity labels from input texts. (ii) **ChatIE** [49] transforms the zero-shot NER task into a multi-turn question-answering problem using a two-stage framework. (iii) **Decomposed-QA** [54] breaks down the zero-shot NER task into a series of simpler sub-problems by labels and follows a decomposed-question-answering paradigm, where the model extracts entities of only one label at a time. (iv) Based on Decomposed-QA, **SALLM** [54] further adopts syntactic augmentation to stimulate the model's intermediate reasoning in two ways, including syntactic prompting and tool augmentation. (v) **SILLM** [55] applies a self-improving framework, which uses unlabeled corpora to stimulate the self-learning abilities of LLMs in zero-shot NER.

In our experiments, we evaluate all the aforementioned baselines for the zero-shot NER task. For few-shot NER, we assess SILLM [55] and Vanilla [54, 55] since ChatIE [49], Decomposed-QA [54], and SALLM [54] do not incorporate task demonstrations in their prompt templates. Additionally, we report the highest results obtained from the model variants of SALLM [54] and SILLM [55] in our experiments. For fair comparisons and cost savings, we employ GPT-3.5 (specifically, the `gpt-3.5-turbo-0125` model[4]) as the LLM backbone for all baselines and agents in CMAS. We use the `text-embedding-ada-002` model,[5] a text-embedding model from OpenAI, to obtain sentence representations. We access OpenAI models using the official API.

### 5.1 Evaluation metrics

Following previous work [54, 55], we conduct our evaluation of zero-shot and few-shot NER tasks using only complete matching and employing the micro F1-score to assess the NER task. We consider a prediction correct only when both the boundary and the type of the predicted entity exactly match those of the true entity.

### 5.2 Implementation details

Following Xie et al. [55] and Wang et al. [42], we set the number of nearest neighbors $K = 50$ during self-annotation and the number of task demonstrations $k = 16$. For self-consistency scores, we set the temperature to 0.7 and sample 5 answers. We set the frequency ratio hyperparameter $\rho$ to 3 for all experiments and only consider 1-gram texts for simplicity.

### 6 Experimental Results

To answer RQ1 and RQ2, we assess the performance of CMAS on both zero-shot and few-shot NER tasks.

### 6.1 Results on zero-shot NER

We turn to RQ1. Table 1 shows the experimental results on both general-domain and domain-specific datasets. To ensure fair comparisons, we reproduce all the baseline models using the `gpt-3.5-`

---

[4]https://platform.openai.com/docs/models/gpt-3-5-turbo
[5]https://platform.openai.com/docs/models/embeddings

`turbo-0125` model, as they are implemented with different versions of GPT-3.5 in the original papers.

We have the following observations: (i) Zero-shot NER is challenging, and baseline models struggle to achieve an F1-score above 60% on most of the datasets. For instance, ChatIE only obtains F1-scores of 37.46% and 29.00% on the WNUT-17 and OntoNotes datasets, respectively. (ii) CMAS achieves the highest F1-scores across all datasets, indicating its superior performance. For instance, CMAS attains F1-scores of 76.23% and 60.51% on the WikiGold and BioNLP11 datasets, respectively. (iii) CMAS significantly outperforms the previous state-of-the-art baselines across all datasets. For example, CMAS achieves improvements of 13.21% and 4.49% over the best-performing baselines on the WNUT-17 and GENIA datasets, respectively.

In summary, CMAS demonstrates its effectiveness in recognizing named entities in a strict zero-shot setting. The identification of contextual correlations and the evaluation of helpfulness scores for task demonstrations are beneficial for zero-shot NER.

### 6.2 Results on few-shot NER

To investigate the effectiveness of CMAS on the few-shot setting, we turn to RQ2. In our experiments, we evaluate CMAS and the baselines in 0-shot, 3-shot, 5-shot, and 10-shot settings, where micro F1-scores are reported. ChatIE, Decomposed-QA, and SALLM are excluded because incorporating gold demonstrations into their prompt templates is non-trivial and beyond the scope of this paper.

Based on Figure 3, we arrive at the following conclusions: (i) Increasing the number of demonstrations with gold labels does not necessarily enhance the prediction performance of LLMs. For example, as the number of demonstrations with gold labels increases from 0 to 10, the F1-score of the Vanilla model significantly drops from 40.10% to 27.10% on the WNUT-17 dataset. This decline may be due to the random selection of demonstrations, which can be highly irrelevant to the target sentence and severely misguide the inference process of LLMs. (ii) CMAS achieves the highest F1-scores and consistently outperforms the state-of-the-art baselines across all few-shot settings, demonstrating its effectiveness and robustness. For example, CMAS exhibits an average improvement of 19.51% and 13.10% over SILLM on the WNUT-17 and GENIA datasets, respectively.

In summary, our proposed CMAS not only effectively extracts entities in the strict zero-shot setting but also achieves the highest F1-scores across all few-shot settings while maintaining robustness to irrelevant demonstrations.

### 7 Analysis

Now that we have addressed our research questions, we take a closer look at CMAS to analyze its performance and generalizability. We examine the contributions of the type-related feature extractor and the demonstration discriminator to its effectiveness (see Section 7.1), investigate its generalizability to different LLM backbones (see Section 7.2) and varying numbers of task demonstrations (see Section A.3), and assess its capability in error correction (see Section 7.3).

**Table 1: Zero-shot NER results (F1) on both general-domain and domain-specific datasets. Numbers in bold are the highest results for the corresponding dataset, while numbers underlined represent the second-best results. Significant improvements against the best-performing baseline for each dataset are marked with ∗ (t-test, $p < 0.05$).**

| Model | Datasets | | | | | |
|---|---|---|---|---|---|---|
| | CoNLL03 | WikiGold | WNUT-17 | OntoNotes | GENIA | BioNLP11 |
| Vanilla [54, 55] | 72.54 | 74.27 | 40.10 | 45.09 | 43.47 | 53.92 |
| ChatIE [49] | 50.13 | 56.78 | 37.46 | 29.00 | 47.85 | 45.56 |
| Decomposed-QA [54] | 52.61 | 64.05 | 42.38 | 35.96 | 34.03 | 57.26 |
| SALLM [54] | 68.97 | 72.14 | 38.66 | 44.53 | 42.33 | 55.06 |
| SILLM [55] | 72.96 | 72.72 | 41.65 | 45.34 | 45.66 | 44.99 |
| CMAS (ours) | **76.43**∗ | **76.23**∗ | **47.98**∗ | **46.23**∗ | **50.00**∗ | **60.51**∗ |

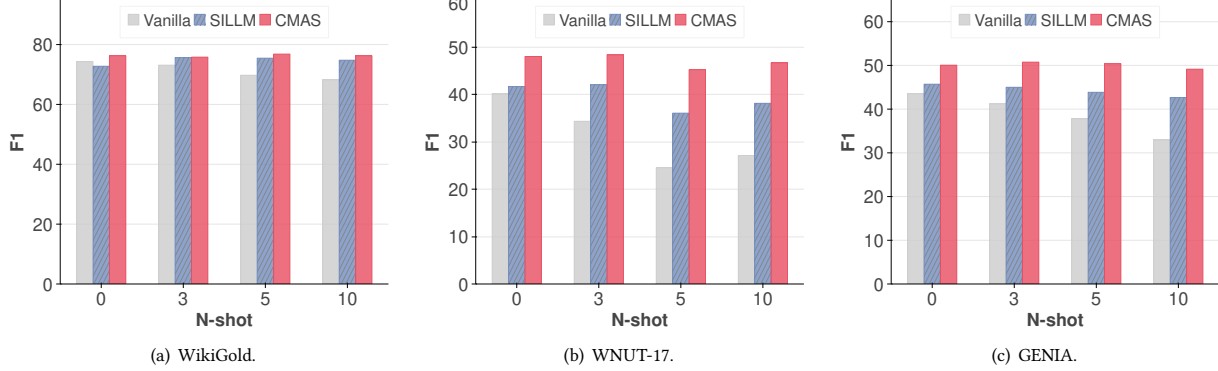

(a) WikiGold.  (b) WNUT-17.  (c) GENIA.

**Figure 3: Few-shot NER results (F1) on WikiGold, WNUT-17, and GENIA.**

## 7.1 Ablation studies

To study the individual contributions of each component to CMAS's performance, we conduct ablation studies on the WikiGold, WNUT-17, and GENIA datasets. The results are presented in Table 2.

Given that the demonstration discriminator relies on entity type-related information from the TRF extractor, it is not feasible to independently remove the TRF extractor. When we ablate only the demonstration discriminator ('- Discriminator'), the overall predictor incorporates only TRF for retrieved demonstrations and target sentences. This exclusion results in a significant drop in CMAS's performance across all three datasets. For instance, CMAS achieves 3.34% and 5.59% higher F1-scores on the WikiGold and GENIA datasets, respectively, compared to its model variant without the demonstration discriminator. These findings highlight the crucial role of evaluating the usefulness of retrieved demonstrations in making predictions. In scenarios where both the demonstration discriminator and the TRF extractor are ablated ('- TRF Extractor'), CMAS reverts to the baseline model, SILLM. The results indicate that identifying contextual correlations surrounding entities considerably enhances SILLM's performance.

In summary, both the demonstration discriminator and the TRF extractor contribute markedly to CMAS's performance improvements over the baselines in the zero-shot NER task.

Furthermore, similar to SALLM, CMAS is readily adaptable for augmentation with external syntactic tools. Following Xie et al.

[54], we obtain four types of syntactic information (i.e., word segmentation, POS tags, constituency trees, and dependency trees) via a parsing tool [10] and integrate the syntactic information into the overall predictor of CMAS using a combination of tool augmentation and syntactic prompting strategies. As shown in Table 2, the inclusion of dependency tree information improves CMAS's performance by 2.52% and 2.94% on WNUT-17 and GENIA, respectively. These results demonstrate that the integration of appropriate external tools further enhances the performance of CMAS.

## 7.2 Influence of different LLM backbones

To explore the impact of different LLM backbones, we evaluate CMAS and baseline models using the latest LLMs, including GPT (gpt-3.5-turbo-0125), Llama (Meta-Llama-3-8B-Instruct[6]), and Qwen (Qwen2.5-7B-Instruct[7]). Table 3 illustrates the zero-shot NER performance on the WNUT-17 and GENIA datasets. We exclude the performance of ChatIE and Decomposed-QA, as their F1-scores with Qwen and Llama backbones are considerably lower than other baselines. As Table 3 shows, CMAS achieves the highest F1-scores when using GPT as the backbone model. Additionally, CMAS consistently outperforms the baselines across various LLM backbones, demonstrating its superiority and generalizability.

---

[6]https://huggingface.co/meta-llama/Meta-Llama-3-8B-Instruct
[7]https://huggingface.co/Qwen/Qwen2.5-7B-Instruct

**Table 2: Ablation studies (F1) on WikiGold, WNUT-17, and GENIA.**

| Model | Datasets | | |
|---|---|---|---|
| | WikiGold | WNUT-17 | GENIA |
| Vanilla [54, 55] | 74.27 | 40.10 | 43.47 |
| ChatIE [49] | 56.78 | 37.46 | 47.85 |
| Decomposed-QA [54] | 64.05 | 42.38 | 34.03 |
| SALLM [54] | 72.14 | 38.66 | 42.33 |
| CMAS (ours) | **76.23** | **47.98** | **50.00** |
| - Discriminator | 73.76 | 45.44 | 48.41 |
| - TRF extractor | 72.72 | 41.65 | 45.66 |
| **External tool augmentation** | | | |
| Word segmentation | **76.92** | 47.63 | 49.22 |
| POS tag | 76.14 | 48.11 | 49.76 |
| Constituency tree | 75.71 | 47.44 | 49.64 |
| Dependency tree | 76.27 | **49.19** | **51.47** |

**Table 3: Influence of different LLM backbones (F1) on WNUT-17 and GENIA. Numbers in bold are the highest results for the corresponding dataset, while numbers underlined represent the second-best results. Significant improvements against the best-performing baseline for each dataset are marked with ∗ (t-test, $p < 0.05$).**

| Model | WNUT-17 | | | GENIA | | |
|---|---|---|---|---|---|---|
| | GPT | Llama | Qwen | GPT | Llama | Qwen |
| Vanilla [54, 55] | 40.10 | 34.88 | 34.93 | 43.47 | 15.36 | 9.97 |
| SALLM [54] | 38.66 | 40.95 | 41.50 | 42.33 | 36.23 | 19.13 |
| SILLM [55] | 41.65 | 22.43 | 36.23 | 45.66 | 28.13 | 33.80 |
| CMAS (ours) | 47.98* | 42.36* | 44.62* | 50.00* | 45.68* | 36.12* |

**Table 4: Numbers of different error types on GENIA. Numbers in bold denote the best results for the corresponding error type, i.e., the least errors, while numbers underlined represent the second-best results.**

| Error Types | | ChatIE | SALLM | SILLM | CMAS |
|---|---|---|---|---|---|
| Type | OOD types | 91 | 2 | **0** | **0** |
| | Wrong types | 52 | 52 | 123 | **32** |
| Boundary | Contain gold | 34 | 7 | 10 | **3** |
| | Contained by gold | **42** | 101 | 240 | 50 |
| | Overlap with gold | 145 | 135 | 318 | **72** |
| Completely-Os | | 122 | 109 | 241 | **59** |
| OOD mentions | | 2 | **0** | **0** | **0** |
| Omitted mentions | | 422 | 342 | 1,109 | **303** |
| Total | | 910 | 748 | 2,041 | **519** |

## 7.3 Error analysis

To investigate CMAS's error correction capabilities, we conduct an analysis of the following errors on the WNUT-17 dataset:

- **Type errors**: (i) **OOD types** are predicted entity types not in the predefined label set; (ii) **Wrong types** are predicted entity types incorrect but in the predefined label set.
- **Boundary errors**: (i) **Contain gold** are incorrectly predicted mentions that contain gold mentions; (ii) **Contained by gold** are incorrectly predicted mentions that are contained by gold mentions; (iii) **Overlap with gold** are incorrectly predicted mentions that do not fit the above situations but still overlap with gold mentions.
- **Completely-Os** are incorrectly predicted mentions that do not coincide with any of the three boundary situations associated with gold mentions.
- **OOD mentions** are predicted mentions that do not appear in the input text.
- **Omitted mentions** are entity mentions that models fail to identify.

Figure 5 (in the Appendix) visualizes the percentages of error types. The majority error types are *overlap with gold* and *ommited mentions*, which account for 72.30% of all errors. These errors may result from incomplete annotations or predictions influenced by the prior knowledge of LLMs. Table 4 summarizes the statistics of error types. With the implementation of the proposed type-related feature extractor and demonstration discriminator, CMAS significantly reduces the total number of errors by 30.60% and 74.60% compared to state-of-the-art baselines SALLM and SILLM, respectively, demonstrating its remarkable effectiveness in error correction.

## 8 Conclusions

We have focused on named entity recognition in the strict zero-shot setting, where no annotated data is available. Previous approaches to zero-shot named entity recognition still encounter two significant challenges: they often overlook contextual correlations and use task demonstrations indiscriminately, both of which can significantly impede the inference process. To tackle these challenges, we have introduced a new framework, *cooperative multi-agent system* (CMAS), which uses the collective intelligence and specialized capabilities of agents. CMAS explicitly captures correlations between contexts surrounding entities by reformulating named entity recognition into two subtasks: recognizing named entities and identifying entity type-related features within the target sentence.

To evaluate the quality of demonstrations, a demonstration discriminator is established to incorporate a self-reflection mechanism, automatically evaluating helpfulness scores for the target sentence and enabling controlled use of demonstrations. Experimental results show that CMAS significantly enhances zero-shot NER performance across six benchmarks, spanning both domain-specific and general-domain datasets. and exhibits strong capabilities in correcting various types of errors.

A limitation of CMAS is that it primarily focuses on a limited set of predefined entity labels. Expanding it to support open NER tasks would be valuable. Future work also includes developing interactive prompt designs, such as multi-turn question answering, for LLM-based agents to iteratively refine or assess responses.

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

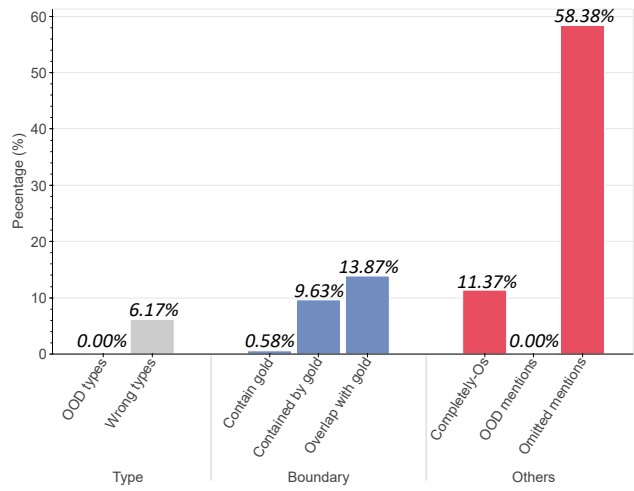

Figure 5: Percentages of different error types on GENIA using CMAS.

## A  Appendix

### A.1  Prompts used in CMAS

Prompt 1, 2, 3, and 4 show prompts used in the self-annotator, TRF extractor, demonstration discriminator, and overall predictor, respectively.

### A.2  Statistics of the datasets used

Table 5 demonstrates the detailed statistics of the dataset used in our experiments.

### A.3  Influence of task demonstration amount

To assess the influence of the number of task demonstrations, we evaluate CMAS and SILLM with varying numbers of task demonstrations, ranging from 2 to 20. Figure 4 details the zero-shot NER

performance on the WNUT-17 and GENIA datasets. It is important

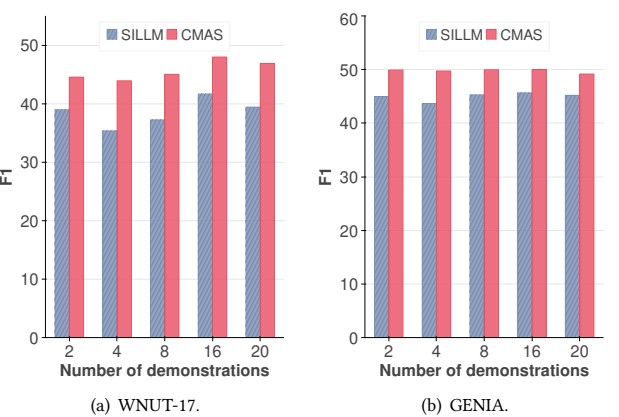

(a) WNUT-17.     (b) GENIA.

Figure 4: Influence of task demonstration amount (F1) on WNUT-17 and GENIA.

to note that other baselines are excluded from this analysis as they do not incorporate task demonstrations in their prompt templates. The results in Figure 4 indicate that CMAS consistently outperforms SILLM across various numbers of task demonstrations on both datasets. Specifically, CMAS achieves an average F1-score improvement of 18.70% and 10.72% over SILLM on the WNUT-17 and GENIA datasets, respectively. This is, CMAS is able to effectively discriminate against irrelevant task demonstrations, maintaining robustness across different numbers of task demonstrations.

### A.4  Percentages of different error types

Figure 5 presents the percentages of error types on GENIA using CMAS.

**Prompt 1: Prompts used for the self-annotator.**

Prompts used for the self-annotator

Given entity label set: ['Organization', 'Person', 'Location', 'Miscellaneous'], please recognize the named entities in the given text. Provide the answer in the following JSON format: [{'Entity Name': 'Entity Label'}]. If there is no entity in the text, return the following empty list: [].

Text: The album cover's artwork was provided by Mattias Noren.
Answer:

**Prompt 2: Prompts used for the TRF extractor.**

Prompts used for the TRF extractor

Here, we provide some example sentences with the corresponding TRFs. TRFs mean type-related features, which are tokens that are strongly associated with the
entity types and relevant to these sentences.

Given entity label set: ['Organization', 'Person', 'Location', 'Miscellaneous'], please identify the TRFs for the target sentences. Provide the answer in the following list format: ['TRF1', 'TRF2', … ].

Text: His parents were encouraged by a friend to develop the child's musical talents and he studied classical piano in the United States.
TRF set: ['father', 'songs', 'player', 'United States', 'French']
……
(Selected demonstrations with pseudo TRF labels)
……
Target sentence: UK Edition came with the OSC-DIS video, and most of the tracks were re-engineered.
TRF set:

**Prompt 3: Prompts used for the demonstration discriminator.**

Prompts used for the demonstration discriminator

Here, we provide some example sentences and the corresponding entity labels and TRFs. TRFs mean type-related features, which are tokens that are strongly associated with the entity types and relevant to these sentences.

Given entity label set: ['Organization', 'Person', 'Location', 'Miscellaneous'], target sentence: 'UK Edition came with the OSC-DIS video, and most of the tracks were re-engineered.' and its TRF set:['video', 'tracks'].

Please predict the helpfulness scores of each sentence, which indicates the degree to which providing the current example can aid in extracting named entities from the target sentence. The score ranges from 1 to 5, with 1 being the least helpful and 5 being the most helpful. Provide answer in the following JSON format: [{'sentence id': 'helpfulness score'}].

Text: His parents were encouraged by a friend to develop the child's musical talents and he studied classical piano in the United States.
TRF set: ['father', 'songs', 'player', 'United States', 'French']
Entity labels: [{'United States': 'Location'}]
……
(Selected demonstrations with pseudo TRF labels and entity labels)
……
Answer:

**Table 5: Statistics of the datasets used. The training set is formed by combining the original training and development sets.**

| Dataset | CoNLL03 | WikiGold | WNUT-17 | OntoNotes | GENIA | BioNLP11 |
|---------|---------|----------|---------|-----------|-------|----------|
| #Train  | 14,382  | 1,422    | 4,403   | 68,452    | 16,692| 3,217    |
| #Test   | 3,453   | 274      | 1,287   | 8,262     | 1,854 | 1,961    |

**Prompt 4: Prompts used for the overall predictor.**

Prompts used for the overall predictor

······

(The corresponding prompt for the demonstration discriminator and its response)

······

Given entity label set: ['Organization', 'Person', 'Location', 'Miscellaneous'], please consider the TRFs and helpfulness scores for the above sentences to recognize the named entities in the target sentence. Provide the answer in the following JSON format: [{ 'Entity Name': 'Entity Label'}]. If there is no entity in the text, return the following empty list: [].

Target sentence: UK Edition came with the OSC-DIS video, and most of the tracks were re-engineered.

Answer:

