# OpenReview forum: "A Cooperative Multi-Agent Framework for Zero-Shot Named Entity Recognition"
_ACM.org/TheWebConf/2025/Conference — WWW 2025 Poster_

### Official Review · Reviewer_HkrK · 2024-11-29

**Novelty:** 5
**Technical Quality:** 6

**Review:**

The paper introduces CMAS (Cooperative Multi-Agent System), a novel framework for zero-shot Named Entity Recognition. CMAS leverages the collaborative abilities of four different agents including a self-annotator, a type-related feature extractor (TRF), a demonstration discriminator, and an overall predictor. The approach addresses two main challenges: not including contextual correlations surrounding entities and the lack of discrimination when using task demonstrations. By redefining NER into two subtasks and using a self-reflection mechanism, CMAS reportedly enhances zero-shot and few-shot NER performance. Experiments are done on six benchmarks outperformingg existing methods.

The main positive points of this work are the following:
1. Innovative Multi-Agent Design. The use of specialised agents with distinct roles improves the effectiveness and adaptability of the system;
2. Performance Gains and use of multiple benchmarks/ CMAS achieves state-of-the-art results on multiple benchmarks, demonstrating its robustness and superiority over existing frameworks.
3. Error Correction.By incorporating type-related features and a self-reflection mechanism, CMAS manages to reduces common errors;
4. TRF extraction -- an interesting, relatively novel and promising task to explore; I think that its formalisation as a standalone task is an important plus of the paper

Some of the weaknesses I identified
The framework currently focuses on a predefined set of entity labels, which restricts its application in open or highly dynamic NER tasks;
The reliance on computationally intensive LLMs like GPT-3.5 could limit accessibility for resource-constrained environments;
The multi-agent architecture, while effective, adds layers of complexity that may pose challenges for implementation and scalability in practical applications;


Minor:
- Please shorten and fit the abstract to one paragraph
- The introduction is also quite lengthy and would benefit from shortening
- Although a general positioning is given, the related work section would benefit from a specific positioning of the proposed approach wrt each of the subsections
- Sections size balance: section 3 is pretty short although important, think of merging
- I would give the research questions earlier (in the introduction) on, not in the experimental section

**Questions:**

I would like to know how the authors would address the several weak points cited above?

**Reviewer Confidence:**

2: The reviewer is willing to defend the evaluation, but it is likely that the reviewer did not understand parts of the paper

**Scope:**

4: The work is relevant to the Web and to the track, and is of broad interest to the community

---

### Official Review · Reviewer_ukEa · 2024-12-03

**Novelty:** 5
**Technical Quality:** 5

**Review:**

### Quality

- The paper tackles the problem of zero-shot NER, the task of recognizing entities without any annotated data, which is a significant problem in the field of NLP.

- The proposed CMAS  framework combines multiple agents to address the outlined challenges, indicating a high level of technical quality. The experiments are conducted across six benchmarks, including both domain-specific and general-domain scenarios, which speaks to the robustness of the evaluation.

### Clarity

- The document is well-structured, with clear sections that guide the reader through the introduction, related work, task formulation, system description, experiments, and conclusions.

- The use of figures and tables helps to illustrate complex concepts and results, enhancing the clarity of the presentation.
The abstract provides a concise overview of the work, its challenges, and the proposed solutions.

### Originality

- The CMAS framework is original in its approach to zero-shot NER by redefining it into two subtasks ((recognizing named entities and identifying entity type-related features) and utilizing a cooperative multi-agent system to capture contextual correlations and evaluate the quality of demonstrations.

### Significance

- The paper addresses a significant challenge in NER, which is crucial for various downstream NLP applications. The proposed solutions have the potential to improve the adaptability and generalizability of NER models, which is particularly important in real-world scenarios where annotated data is scarce.

- The work demonstrates state-of-the-art performance on multiple benchmarks, which is a significant contribution to the field.

### Pros

- Re-define the NER task to address the critical problem with a novel multi-agent approach.

- Demonstrates effectiveness in both zero-shot and few-shot settings.

### Cons

- The application of a multi-agent system may require significant computational resources, yet the paper lacks relevant analysis with other state-of-the-art models in terms of computational efficiency.

- The multi-agent system seems too simple compared to many existing systems for other tasks. Therefore, the improvements may rely on the LLMs themselves or the prompts' design.

- The paper does not extensively discuss the potential scalability issues or the implications of using multiple agents in different contexts.

**Questions:**

- The reliance on LLMs may introduce biases present in the pre-training data. Have the authors conducted analyses on different models to watch their adaption ability on the NER task?

- What are your principles of prompt designing to ensure the multi-agents system's success rate and performance?

- Can the shortcomings mentioned in the error analysis be addressed through refining the design of the system or prompts?

**Reviewer Confidence:**

3: The reviewer is confident but not certain that the evaluation is correct

**Scope:**

3: The work is somewhat relevant to the Web and to the track, and is of narrow interest to a sub-community

---

### Official Review · Reviewer_TKZu · 2024-12-03

**Novelty:** 3
**Technical Quality:** 4

**Review:**

The manuscript proposes CMAS for addressing challenges in zero-shot Named Entity Recognition (NER) by leveraging multiple specialized agents.

**Questions:**

1. The approach involves multiple computational steps, including mutual information computations and self-consistency mechanisms, potentially increasing inference costs.  while, authors do not discuss the computational complexity, which is important for practical applications.
2. CMAS is limited to a fixed set of predefined entity types, which restricts its adaptability for open NER tasks.

**Reviewer Confidence:**

3: The reviewer is confident but not certain that the evaluation is correct

**Scope:**

3: The work is somewhat relevant to the Web and to the track, and is of narrow interest to a sub-community